

# The conservation value of freshwater habitats for frog communities of lowland fynbos

Naas Terblanche[1] and John Measey[2]

[1] Riverglade Retirement Village, Parklands, Unaffiliated, Cape Town, South Africa
[2] Centre for Invasion Biology, Department of Botany & Zoology, Stellenbosch University, Stellenbosch, South Africa

## ABSTRACT

Amphibians are more threatened than any other vertebrate class, yet evidence for many threats is missing. The Cape lowland fynbos (endemic scrub biome) is threatened by habitat loss, and natural temporary freshwater habitats are removed in favour of permanent impoundments. In this study, we determine amphibian assemblages across different freshwater habitat types with special attention to the presence of invasive fish. We find that anuran communities differ primarily by habitat type, with permanent water habitats having more widespread taxa, while temporary water bodies have more range restricted taxa. Invasive fish are found to have a significant impact on frogs with toads most tolerant of their presence. Temporary freshwater habitats are a conservation priority in the area, and their amphibian assemblages represent endemic taxa that are intolerant of invasive fish. Conservation of a biodiverse amphibian assemblage in lowland fynbos areas will rely on the creation of temporary freshwater habitats, rather than a northern hemisphere pond based solution.

## INTRODUCTION

Amphibian conservation has centred around three major themes: habitat change, disease and invasive species (*Grant et al., 2019*). Freshwater habitats are in particular peril, with declines that are far greater than terrestrial ecosystems (*Dudgeon et al., 2006*). However, the construction of artificial impoundments is increasing rapidly (*Downing et al., 2006*), and changing the nature of landscapes especially in arid ecosystems, where these permanent impoundments facilitate invasions of freshwater species (*Johnson, Olden & Vander Zanden, 2008*). Habitat change and invasive species impact more species globally and are the proximate causes of conservation concern for the majority of amphibian species (*Harfoot et al., 2021*). Despite a general acknowledgement of these mechanisms, conservation evidence for impacts and their commensurate measures for the conservation of amphibians continues to be low (*Meredith, Van Buren & Antwis, 2016*).

The impacts of invasive species on amphibians have been assessed qualitatively (*Bucciarelli et al., 2014*; *Falaschi et al., 2020*), and quantitatively (*Nunes et al., 2019*). Invasive freshwater fish are ranked highly in reviews of invasive species as causing severe

Corresponding author
John Measey,
johnmeasey@gmail.com

impacts on many amphibian communities (*Hecnar & M'Closkey, 1997*; *Ficetola & De Bernardi, 2004*; *Hartel et al., 2007*; *Holbrook & Dorn, 2016*). Excluding invasive fish from sites with threatened frog species has resulted in recovery of anuran populations in Spain and Portugal (*Rana iberica Bosch et al., 2019*) and California (*Rana mucosa Knapp, Boiano & Vredenburg, 2007*), leading those workers to identify the proximate role of invasive fish as a threat to amphibian populations. But the impacts of invasive fish are poorly described in the southern hemisphere (except Australia), especially with respect to amphibian communities. However, many amphibian communities are driven by natural environmental factors as well as anthropogenically driven creation and modifications of freshwater habitats (*Ficetola & De Bernardi, 2004*; *Hartel et al., 2007*; *Kruger, Hamer & Du Preez, 2015*).

The low-lying fynbos of South Africa's Cape region is a biodiversity hotspot (*Myers et al., 2000*), composed of evergreen, Mediterranean scrub vegetation (*Mucina & Rutherford, 2006*). The fynbos also holds an important community of amphibians that have high conservation concern (*Measey, 2011*; *Schreiner, Rödder & Measey, 2013*; *Mokhatla, Rödder & Measey, 2015*). Much of the habitat where amphibians and other flora and fauna were once abundant has been transformed for agriculture and more recently for housing (*Measey & Tolley, 2011*; *Rebelo et al., 2011*; *Measey et al., 2014*). Where land has been transformed, temporary wetlands have been infilled and permanent impoundments (dams) or ponds added to the landscape. The addition of permanent water and the introduction of alien fish has been ongoing for ~200 years (*Ellender & Weyl, 2014*). Angling is a popular pastime in the region, and anglers introduce fish to new impoundments and natural waterbodies (*Ellender et al., 2014*).

Southern Africa has no salamanders or caecilians, but several major radiations of anurans, many of which specialise in lowland temporary aquatic habitats (*Poynton, 1964*). The extreme southwestern corner of the continent has a mediterranean climate with winter rains and dry hot summers (*Wilson et al., 2020*). Several species that rely on temporary water have become threatened, while those that thrive in permanent water have become abundant and ubiquitous even in arid areas. Examples of IUCN threatened species in the area include the Western Leopard Toad *Sclerophrys pantherina* (EN), the Cape Platanna *Xenopus gilli* (EN), the Microfrog *Microbatracella capensis* (CR), and the Flat Caco *Cacosternum platys* (NT).

In this article, we aim to determine whether habitat characteristics (especially temporary *vs* permanent water) correlate with Cape lowland amphibian communities. In particular, we were interested to find out whether anthropogenically constructed impoundments are useful sites for threatened amphibian communities, in the presence or absence of invasive fish.

## METHODS AND MATERIALS

### Site selection

Using Google Earth imagery from 2017, we classified every waterbody visible within our study area (two catchments) into our nominate freshwater body types being natural: vleis (natural temporary shallow water bodies), natural pools and river edges in the fynbos, and

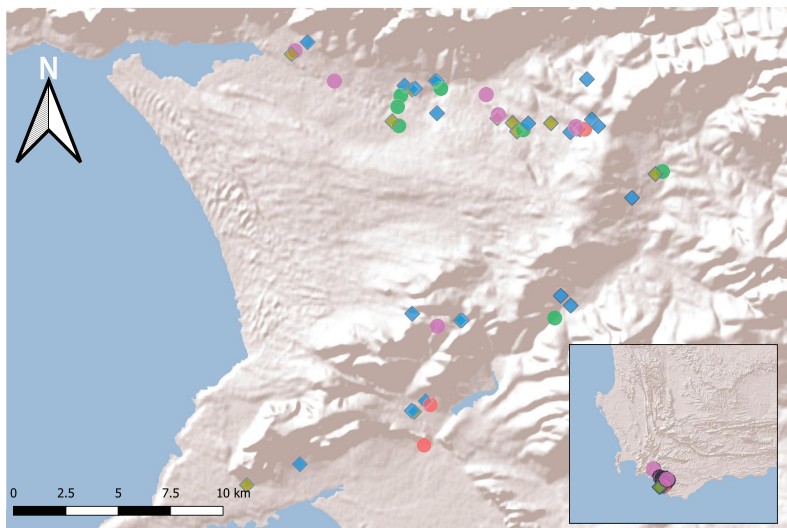

**Figure 1** **Fifty sampling sites in the Overberg region of South Africa.** Freshwater bodies (coloured by wetland type: Temporary vlei purple, River edge green, Large dam brown, Small dam blue and Fynbos pool red) are constructed (diamonds) or natural (circles) (inset shows extreme southwest of southern Africa). For details of the selected sites see Table S1.   

anthropogenically created: small dams (artificial impoundments <2,000 m² including ponds), and large dams (artificial impoundments >2,000 m²). In lowland fynbos, slow moving rivers are effectively temporary water bodies and have the same amphibian species assemblages (*Channing, 2019*). This gave us a candidate list of 196 sites (see Table S1) all chosen from within the fynbos biome (see *Mucina & Rutherford, 2006*).

We made our initial stratified sampling selection from within these 196 sites to represent balanced numbers of freshwater body types, equally represented across space, and these were further refined once we requested permission to access sites from landowners. The final 50 sites selected fell within two separate catchments, varied in their spatial proximity and different water body types (see Fig. 1; Supplemental Information; Table S1).

## Anuran data collection

Each site was visited three times during the day and into the first half of the night in the active winter period (between May and August 2016–2017), with a pseudorandom order of sites as permission to visit was granted. We changed the order of the visits to sites between years to attempt to visit each site both early and late in the austral winter rainy season, and therefore maximise detection of speces that are active at different periods of the winter. At each site we performed a standardised method to survey anurans. Firstly, we walked around the waterbody during the day and again at night with a headlight looking for adult anurans. We set audio recorders at each site for two nights on each site visit to collect calling data. Lastly, we set funnel traps over two nights for tadpoles and adult aquatic frogs (*Xenopus* species). After identification, all individuals were immediately released on site. All fieldwork was authorised by CapeNature (permit number: AAA043-00449).

The research protocol was approved by Stellenbosch University Research Ethics Committee: Animal Care and Use (ethics number: SU-ACUD15-00101).

We identified calls using spectrograms in Audacity (http://audacityteam.org/) against a set of calls for species in South Africa (*Du Preez & Carruthers, 2017*). Adults were identified against descriptions and keys in two field guides (*Du Preez & Carruthers, 2017*; *Channing, 2019*). Tadpoles were identified from their mouthparts according to *Du Preez & Carruthers (2017)* using a stereomicroscope for exemplars from each site. Taxonomy for all species was corrected to *Frost (2020)*, and we consulted relevant new literature with respect to newly described cryptic species. For example, the genus of Dainty Frogs, *Cacosternum*, was found to have multiple cryptic species by *Channing et al. (2013)*, but only one of these, *C. australis*, has been identified within this area (see *Vogt et al., 2017*).

## Fish data collection

To determine whether invasive fish were present at each locality, we consulted landowners for information on the presence/absence of invasive fish. In this region, invasive fish are stocked for game fishing (with the exception of *Gambusia* spp.), and no native fish are stocked or present in dams (which are all stocked by landowners). In addition, no fish are present in temporary water sites. In general, landowners are very knowledgable about the fish species present in their dams. As a means of validating landowner information, we asked local recreational fishermen to submit images of species caught within dams in our sampling area. In every casae, this corroborated landowner information. In two cases when there was ambiguity (the landowner was unsure whether fish were still present), we sampled the fish in the dam. As we did not sample all sites in a methodological manner, we only use presence/absence of invasive fish in our analyses.

## Site data collection

For each site, we measured the area and perimeter of the waterbody using tools in Google Earth with images from mid-Winter (June and July) when they were at their maximum size, to correspond with our sampling date. We also noted the latitude and longitude of the centre point of each site, and its catchment. Previous studies have been suggested area and perimeter to be of importance in structuring amphibian communities (*Hamer & Parris, 2011*; *Kruger, Hamer & Du Preez, 2015*). We measured conductivity as some sites were in close proximity to the sea, and pH (Hannah Instruments) as low pH has been considered important to species inhabiting naturally acidic fynbos pools (*e.g.*, *Picker, McKenzie & Fielding, 1993*). As vegetation in the fynbos is typically low, we did not record vegetation surrounding aquatic habitats. Hydroperiod of sites was recorded as it is known to be of great importance for amphibian community structure (*Pechmann et al., 1989*; *Wellborn, Skelly & Werner, 2003*). Together with our waterbody class (see site selection above), these measures were used as environmental covariates in the data analysis.

## Data analyses

We first investigated our site covariates for multicollinearity using Variance Inflation Factors (VIF) in the car package (*Fox & Weisberg, 2019*). We considered a VIF threshold

value of <5 as causing negligible collinearity effects. Subsequently, we divided the site environmental variables into two groups: those environmental variables with the potential to structure amphibian communities (pH, conductivity, hydroperiod, area, perimeter and invasive fish), and covariates which might confound the analysis through spatial autocorrelation, catchment assignment or the order of visit (date visited, latitude, longitude and catchment).

To test which of our environmental variables explained the frog community, we ran a partial Redundancy Analysis (partial RDA) in package vegan (*Oksanen et al., 2022*). RDA is a constrained ordination analysis that models the effects of a matrix of explanatory variables (here environmental variables) on a complementary community matrix (here species of amphibians) using multiple regression (*Legendre & Legendre, 2012*). To account for potential spatial autocorrelation when conducting the RDA, we converted our Latitude and Longitude to Moran's Eigenvectors Maps (MEMs) using the dbmem function in package adespatial (*Dray et al., 2023*). MEMs maximize Moran's coefficient of spatial autocorrelation by identifying spatial patterns that are most strongly correlated with the data, and are then used to improve the accuracy of resulting statistical models (*Guénard & Legendre, 2022*; *Dray et al., 2023*). The partial RDA used in this study allows for a third matrix of covariates (here MEMs and date data) to be controlled for when conducting the RDA by calculating the residuals of the two covariate matrices before conducting the RDA (*Legendre & Legendre, 2012*). In effect, this removes the effect of these background variables before the RDA proper. For our frog community data, we used presence/absence data for each of the identified anuran species at each of the 50 sites. Presence was determined through either adults captured, calls recorded or tadpoles in traps.

In our partial RDA model we used frog species as our community matrix (Y), the environmental variables as our explanatory matrix (X), and the spatial and date covariates as the condition matrix (Z) in the formula: rda (Y ~ X + Condition (Z)). We calculated the adjusted $R^2$ of the model to determine the proportion of the percentage of the variance in the amphibian community explained by the variation in frog community composition across sites, and we performed a permutation test with 10,000 steps (using anova.cca in package vegan) to test whether the model is significant. We used further permutation tests (with 10,000 steps each using anova.cca) to test for the significance of each variable and each canonical axis (using "term" and "axis", respectively). Lastly, we used ggplot2 (*Simpson, 2015*) to plot the partial RDA model results.

To further explore the influence of invasive fish and other environmental variables on the anuran community, we used an unconstrained ordination analysis on a reduced dataset with a non-metric multidimensional scaling (NMDS) analysis using metaMDS with the Jaccard similarity index (suitable for presence/absence data) in package vegan with a maximum of 1,000 tries, and no autotransformation. We increased the number of dimensions (k) until increases failed to reduce the stress value by more than 0.05. We then used envfit in package vegan with our reduced set of continuous and discrete environmental variables (see above for removal of correlated environmental variables) to determine whether they were significant determinants of our amphibian species assemblages. We used envfit to discover which of the amphibian species were contributing

**Table 1 The 11 species of amphibians found at 50 lowland sites in the Overberg, South Africa.** The number of sites is provided with the number of temporary sites in brackets. Their position in ordinal space and from a partial RDA analysis demonstrate affinity. Species are sorted according to their position along RDA1 (see Fig. 2). Mean snout-vent length (SVL) is taken from AmphiBIO (*Oliveira et al., 2017*). Range sizes are calculated from Extent of Occurrence from the IUCN RedList (www.iucnredlist.org).

| Species | Family | Number of sites | RDA1[*] | SVL (mm) | IUCN range (km$^2$) |
|---|---|---|---|---|---|
| *Amietia fuscigula* | Pyxiecephalidae | 27 (5) | 0.5049 | 125 | 598,013 |
| *Hyperolius horstocki* | Hyperoliidae | 19 (6) | 0.5148 | 43 | 18,110 |
| *Tomopterna delalandii* | Pyxiecephalidae | 10 (2) | 0.4277 | 41 | 215,909 |
| *Xenopus laevis* | Pipidae | 34 (8) | 0.3862 | 147 | 3,761,124 |
| *Scelerophys pantherina* | Bufonidae | 12 (1) | 0.2792 | 140 | 3,824 |
| *Scelerophys capensis* | Bufonidae | 12 (2) | 0.1671 | 115 | 732,181 |
| *Arthroleptella villiersi* | Pyxiecephalidae | 8 (3) | 0.0588 | 22 | 6,382 |
| *Semnodactylus wealii* | Hyperoliidae | 4 (2) | 0.0687 | 44 | 376,520 |
| *Cacosternum australis* | Pyxiecephalidae | 29 (10) | 0.1595 | 26 | 17,037 |
| *Strongylopus bonaespei* | Pyxiecephalidae | 4 (3) | −0.1525 | 42 | 28,077 |
| *Strongylopus grayii* | Pyxiecephalidae | 39 (13) | −0.2411 | 64 | 580,275 |

**Note:**
[*]Note that RDA 2 is not shown here as, although it is shown in Fig. 2, it was not significant.

most to the community distributions. Because invasive fish can only occur in permanent water, we reduced the dataset to the 36 permanent water sites. We then used a permanova on distance matrices (adonis2) in vegan. We used the coefficients of the output from the permanova to determine amphibian species sensitivity to fish.

## RESULTS

We identified 11 different anuran species (Table 1) across the 50 sites sampled. All sites sampled were found to have at least one species of anuran, with a maximum of nine and a minimum of one. No species were found exclusively at permanent or temporary water sites (Table S2). In addition, we had evidence of three invasive fish species: Large-Mouth Bass *Micropterus salmoides*, Small-Mouth Bass *M. dolomieu* and Mozambique Tilapia *Oreochromis mossambicus* from 14 sites.

The partial RDA model explains 11.06% of the variation in frog community composition across all sites, and is statistically significant (F = 1.583; *P* = 0.002). From the partial RDA, we found that the best measured environmental determinants of the amphibian community were the presence or absence of invasive fish (F = 1.950; *P* = 0.035; Fig. 2; Table 2), area (F = 1.828; *P* = 0.050) and the type of wetland habitat sampled (F = 2.046; *P* < 0.001). In Fig. 2, only the first axis was found to be significant (F = 6.008; *P* < 0.001).

Our exploratory analysis on how invasive fish impact amphibian assemblages in permanent water (using 36 of the 50 sites) produced a 2-dimensional NMDS fit with stress of 0.192, which showed significant differences using adonis2 (R$^2$ = 0.089; *P* = 0.025; Fig. 3). The two species most tolerant of the presence of fish were the toads: *Sclerophrys capensis* and *S. pantherina*, while the species most intolerant of fish was *X. laevis* (Fig. 3; Table S3).

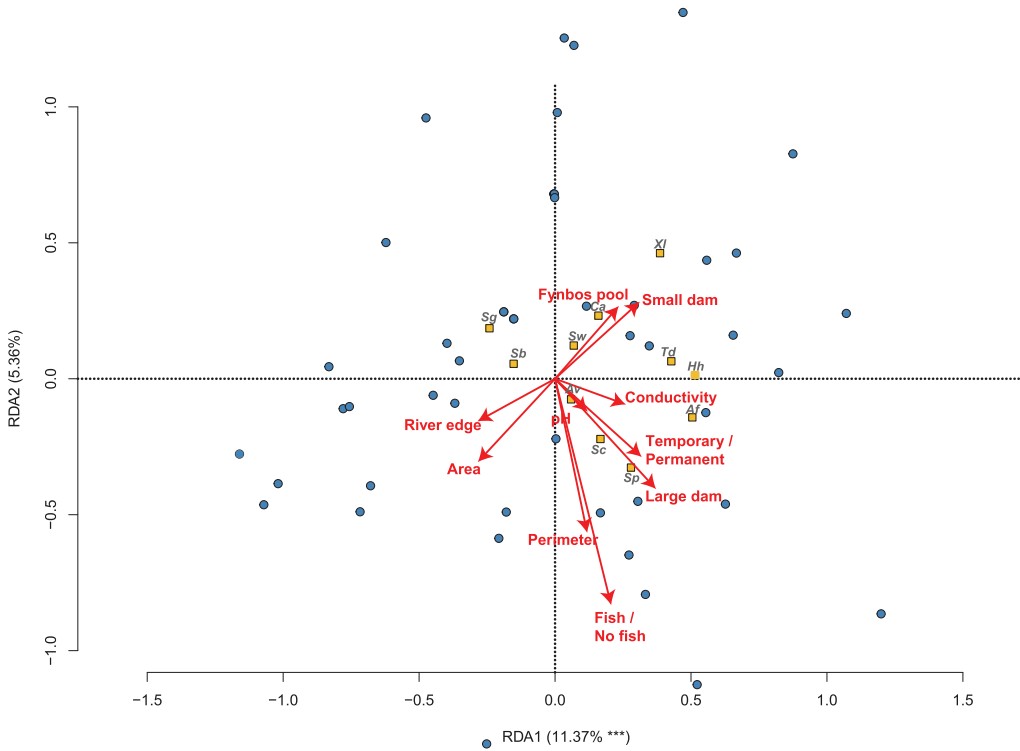

**Figure 2 The relationship between 50 sites sampled and their amphibian communities in the Overberg region of South Africa from a reduced redundancy analysis (reduced RDA).** The position and influence of environmental variables are shown with arrow lengths, when variables are factors separate arrows are shown for each factor. Species names are abbreviated to the first letters of genus and specific name (see Table 1).          

**Table 2 The significance of environmental variables measured in structuring the community of 11 species of amphibians found at 50 lowland sites in the Overberg, South Africa.** Outputs are from a partial Redundancy Analysis (partial RDA) controlling for spatial position of the sites and their day of sampling. Asterisks (* and **) denote levels of statistical significance.

| Environmental variable | df | Variance | F | P |
|---|---|---|---|---|
| Fish/No fish | 1 | 0.0860 | 2.4115 | 0.009** |
| Temporary/Permanent | 1 | 0.0570 | 1.5985 | 0.096 |
| Area | 1 | 0.0500 | 1.4029 | 0.179 |
| Perimeter | 1 | 0.0576 | 1.6141 | 0.107 |
| pH | 1 | 0.0156 | 0.4364 | 0.927 |
| Conductivity | 1 | 0.0504 | 1.4144 | 0.171 |
| Wetland type | 4 | 0.2208 | 1.5476 | 0.033* |
| Residual | 35 | 1.2483 | | |

# DISCUSSION

Our study stresses the importance of freshwater types which determine the type of amphibian community in the southwestern Cape, with an important division between anthropogenically created water bodies (irrespective of size), and those that occur naturally

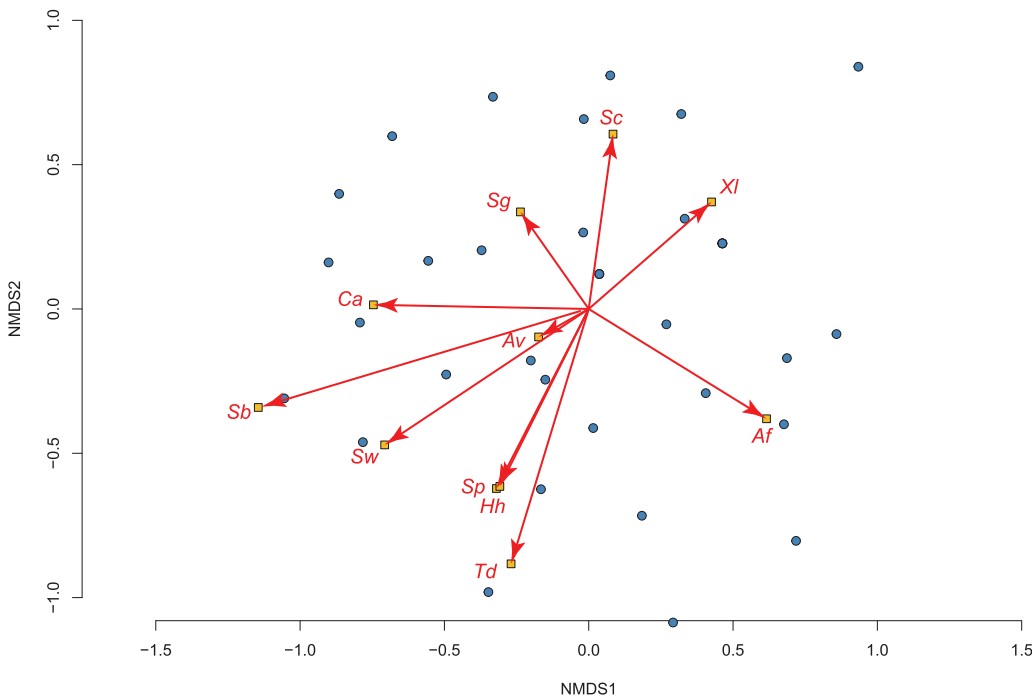

**Figure 3 The relationship between 36 permanent water sites sampled and their amphibian communities in the Overberg region of South Africa using a non-metric multidimensional scaling (NMDS) analysis.** Points and ellipses are coloured by whether fish are present (red) or absent (blue). The position and influence of species are shown with arrow lengths. Species names are abbreviated to the first letters of genus and specific name (see Table 1).     

in the fynbos. Permanent water bodies generally hold widespread species, while temporary sites typically hold fynbos endemic species. In addition, we show that the presence of invasive fish in permanent water bodies also impacts amphibian assemblages. Our results indicate that building permanent water bodies, whether they be large impoundments for agricultural water supply or small garden ponds, will favour different amphibian communities from those present in sites with temporary water. Many urban homeowners create permanent small ponds in their gardens with conservation goals. However, our results indicate that trends for increasing biodiversity in urban areas by creating ponds championed in the northern hemisphere (*Hassall, 2014*; *Hill, Lawson & Tuckett, 2017*; *Hill et al., 2018*) are inappropriate in the fynbos where large impoundments already provide for assemblages that require permanent water. Permanent impoundments also promote invasions of both fish and amphibians (*Davies et al., 2013*; *Ellender et al., 2014*). Currently, amphibians that rely on temporary water in lowland fynbos are poorly served by anthropogenically created wetlands, but could be better conserved by the promoting construction of temporary water bodies instead of ponds.

While no anuran species was exclusive to permanent water, these types of water bodies were commonly associated with more widespread species: *Amietia fuscigula*, *Hyperolius horstocki*, *Xenopus laevis* and *Tomopterna delalandii*. These species are not endemic to the area, while those associated with temporary water have much smaller distributions (~20,000 km$^2$). Toads (*Sclerophrys capensis* and *S. pantherina*) were most tolerant of the

presence of invasive fish, presumably because their eggs and larvae are toxic and adults have prominent parotid glands (*Hecnar & M'Closkey, 1997*; *Crossland & Alford, 1998*; *Caller & Brown, 2013*). The species most intolerant to the presence of fish was *X. laevis*, which may be because they are principally aquatic and encounter fish more often than other frogs. The area we sampled did not include some threatened species present in the lowland fynbos, for example *X. gilli* (EN) and *Microbatrachella capensis* (CR), but these are most commonly associated with temporary water (J. Measey, 2017, Personal Observation).

Historically, these areas of lowland fynbos would have had very few permanent water bodies. The sediment is typically sand or silty soils over young Quaternary sediments, largely derived from weathering Table Mountain sandstones and Cape Supergroup shales (*Cawthra et al., 2020*). Rivers that flow year round may well have been augmented by the movements of large mammals to increase the permanent water features associated with them (*Venter et al., 2020*). Away from rivers, most water bodies would have formed through rainfall, or be fed by underground seepages, during the wet winter period, and completely dry out during summer. Many of the lowland fynbos areas have been developed and habitat loss continues to the present day (*Skowno, Jewitt & Slingsby, 2021*). The Cape Lowland Freshwater Wetlands are considered to be Critically Endangered in the National Ecosystem Status for South Africa (*Dayaram et al., 2021*).

Because we were not able to sample each site systematically for fishes, there is a chance that we have false negative data among some of the permanent water sites. For example, we found no mosquitofish, *Gambusia affinis*, in the water bodies we sampled, but they are known to be invasive in many drainages of South Africa (*Weyl et al., 2020*). Additionally, sites along river edges are likely to have more transient impacts from fish, unlike those of ponds of similar sizes. These false negatives may impact the reported position of anuran species in relation to invasive fish, but they are unlikely to change the overall result. Similarly, we cannot discount the possibility that some anuran species went undetected during our surveys.

We did not include native fish in our scoring. To our knowledge, none of the impoundments that we surveyed contained any native fishes. Sites along the river are reported to have Cape Kurper *Sandelia capensis* and *Galaxias* sp. 'Klein' (see *Chakona, Swartz & Gouws, 2013*). Of these, the Cape Kurper may have exerted some predation impact on amphibians. There are other native predatory species that may exert an impact on amphibian communities, such as the Cape Clawless Otter *Aonyx capensis*, Cape Terrapin *Pelomedusa galeata* and the Western Cape River Crab *Potamonautes perlatus*. All of these species are present in the area sampled and further study would be required to interpret their impact on amphibian communities.

## CONCLUSIONS

Anthropogenically created permanent water bodies (regardless of size) and the presence of invasive fish significantly alter amphibian communities in lowland fynbos by favouring widespread species. Our results question the dogma of creating urban ponds to increase biodiversity (*Hassall, 2014*; *Hill, Lawson & Tuckett, 2017*; *Hill et al., 2018*), at least for amphibian communities but possibly for other species. Recent success in restoring

European amphibian populations with pond construction (*Moor et al., 2022*) needs to be taken in context, and not as a freshwater biodiversity panacea. Rather like the popular fixation on planting trees, the evolutionary and climatic context must take precedence when considering future conservation actions (*Bond et al., 2019*). While our research is pertinent to low-lying areas of the fynbos, the importance of hydroperiod in structuring aquatic communities, including amphibian communities, has been stressed before (*e.g.*, *Pechmann et al., 1989*; *Welborn, Skelly & Werner, 1996*; *Werner et al., 2007*; *Holbrook & Dorn, 2016*). This is even more important in mediterranean biomes where permanent water is an unusual natural feature, but anthropogenic need for access to water particularly for agriculture have made it the most abundant of freshwater aquatic features. It may be that other southern African biomes may also have their amphibian communities strongly impacted by hydroperiod, but this remains untested (*Kruger, Hamer & Du Preez, 2015*). When opportunities arise for mitigation effects that call for creation of wetland habitats in the fynbos, we strongly encourage creation of temporary water features that are allowed to dry out during the summer months. This effectively excludes populations of invasive fish and increases the diversity of amphibian fauna endemic to the southwestern Cape lowlands.

## ACKNOWLEDGEMENTS

We would like to thank all landowners for permission to survey fish and frogs on their property. We would like to thank Andrew Turner and Martine Jordaan from CapeNature for their help and assistance during this study. We thank the editor and reviewers, especially Diogo Provete, for their useful comments that improved the quality of this manuscript.

### Funding

John Measey was supported by the Centre of Excellence for Invasion Biology and Stellenbosch University. The funders had no role in study design, data collection and analysis, decision to publish, or preparation of the manuscript.

### Grant Disclosures

The following grant information was disclosed by the authors:
Centre of Excellence for Invasion Biology and Stellenbosch University.

### Competing Interests

John Measey is an Academic Editor for PeerJ.

### Author Contributions

- Naas Terblanche conceived and designed the experiments, performed the experiments, authored or reviewed drafts of the article, and approved the final draft.

- John Measey conceived and designed the experiments, analyzed the data, prepared figures and/or tables, authored or reviewed drafts of the article, and approved the final draft.

## Animal Ethics

The following information was supplied relating to ethical approvals (*i.e.*, approving body and any reference numbers):

The Stellenbosch University Research Ethics Committee: Animal Care and Use approved the study (SU-ACUD15-00101).

## Field Study Permissions

The following information was supplied relating to field study approvals (*i.e.*, approving body and any reference numbers):

All fieldwork was authorised by CapeNature (permit number AAA043-00449).

## Data Availability

The data is available at OSF: Measey, John, and I L Terblanche. 2022. "Freshwater Habitats for Frog Communities of Lowland Fynbos, South Africa." OSF. November 18. DOI 10.17605/OSF.IO/8HDQE.

## Supplemental Information

Supplemental information for this article can be found online at http://dx.doi.org/10.7717/peerj.15516#supplemental-information.

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
