# Peer review of "The conservation value of freshwater habitats for frog communities of lowland fynbos"

_PeerJ, doi:10.7717/peerj.15516_

## Round 0.1 · original submission · Major Revisions

While Reviewer 1 has noted several weaknesses with your manuscript, the two other reviewers see merit in your paper and recommend it for publication after attending to the reviewers. I agree that your manuscript has potential for publication, but you must comprehensively address the comments raised by Reviewer 1 and the other reviewers. Specifically, pay attention to the questions raised on the methodology and provide justification for the decisions made and their implications on the findings- for instance, on pooling data from lotic and lentic habitats. Similarly, you need to review your data analysis and re-analyze using the statistical approaches suggested by the reviewer (Reviewer 1). I look forward to your revised paper detailing actions taken in response to all the minor and major comments by the reviewers. Also, note that your revised manuscript is thoroughly checked for grammar and clarity for re-submission.

·

Basic reporting

Authors have explain to a wider audience what 'fynbos' mean. Authors keep saying that no study have investigated a number os topics, when actualy there're a number of studies that are missing from references. These are too many for me to list here, but in L. 58-9, look for papers in Austral Ecology, Hydrobiologia, and other herpetological journals. As for L. 44-6, look for papers on a global scale by Rahbek, Araújo, and Diniz-Filho.
You're also missing key literature on the role of hydroperiod gradient on amphibians, such as Wellbordn et al. 1997, several papers by David Skelly, E. Werner, Resetarits etc

Experimental design

Frog communities inhabiting lentic habitats (e.g., ponds) are usualy very different from the ones inhabiting water-flowing habitats (e.g., streams). Thus, I'm not sure if mixing those two types of habitat together if a good idea, especially if your goal is to determine the role of temporary vc permanent water bodies.
How sure can you be from identifying tadpoles in the field (L. 112)? You said that you ID tadpoles from their mouthparts (L. 120), how did you do it without a stereomicroscope?
You can't just rely on indirect evidence for fish presence, why didn't you sample for fish?
You didn't provide ay background on why you think the environmental variables you collect will influence species composition. For example, several papers (e.g., Provete et al. 2014 and ref cited therein) found that water chemistry variables don't influence frog species composition at the lanscape scale.

Validity of the findings

The exploratory data analysis is problematic, at best. For example, you mention you calculated a correlation, but then provide and R2 (L. 141), which is a coefficient calculated in regression. Additionally, a 0.3 threshold is way too low. You'd be better off using a VIF here.
Your decisions on what variables to retain (L. 143-6) look too arbitrary. Which study found that perimeter is more important than area? This is basic species-area relationship.
The statistical analyses are not correct for a number of reasons. But the main one is that, if you want to test the role of environmental variables on species composition you need a constrained, not an unconstrained ordination. An nMDS is unconstrained ordination, which uses an iterative algorithm to find the best position of objects and variables in a pre-determined number of dimentions (usually 2 or 3). It doesn't calculates eigenvectors and eigenvalues. What you did afterwards was simply projecting the environmental variables onto the reduced space created by nMDS, but this procedure is done irrespective of the species composition. So, you can't test hypothesis. On a side not, a STRESS alone is not a good measure of the goodness-of-fit of nMDS, you need a Shepard diagram. The correct way to test the effect of environmental variables on a matrix of species composition is to use a constrained ordination, such as RDA, CCA or even beter, a model-based method, such as gllvm, HMSC.
Also, since you sampled both species environmental variables in space, they certainly exhibit spatial autocorrelation (Legendre 2003). This will certainly influence your coefficients and the inference. However, you haven't incorporated spatial autocorrelation into the analysis, nor even tested for it. From Fig. 1, many sites are right beside each other. They certainly are not independent in terms of both species and environmental variables.

Additional comments

The map in Fig 1 is just too confusing, there're colors and symbols mixed on the top of tiny white triangles.
adonis2 is not the name of the method. This is the function name in vegan that implements a PERMANOVA, and you need to declare the coef of distance used.

Reviewer 2 ·

Basic reporting

Some minor suggestions regarding the figures/tables:

Figure 1
I would recommend moving the brackets at the end referring to Table S1 to the end of the sentence.

Figure 2
In your figure caption you refer to (a), (b), (c), and (d), however, these labels are missing from the figure itself.

Figure 3
In your figure caption you do not need to make reference to “(a)” as there is no secondary component to the figure.

Table 1
In the table caption you say see Figure 1 when referring to the output of envfit. However, Figure 1 is your map image. Perhaps you mean Figure 2 or supplementary material Table S2?
Also, it might be interesting to break up the column regarding the number of sites each species was found in into permanent and non-permanent water bodies (even if its just adding this information in brackets)

Experimental design

I do not have any major issues with the paper's experimental design, but I think some extra clarification or information on some aspects would be beneficial:

1. Line/s: 110-117
Just a point of interest, did you select particular nights/days to survey using either method (aural or visual) to ensure the maximum possible likelihood of detections, e.g. just after rain or at a particular time of day? Also, how intensive was your sampling (i.e. how long did you search/listen)? Was this kept fairly consistent across all 50 sites?

2. Line/s: 129-132
Do you know if smaller fish species that are not actively fished, but are still active and predating on tadpoles, are likely to be present in any of the ponds?
Also, there seems to be a possibility that the sites recorded as absent of fish may be presenting as a false negative as information on fish presence was collected using secondary sources (that may be biased towards larger/easily caught fish). Some acknowledgement of this in the discussion should probably be included.

3. Line/s: 139
You collected data on water pH, salinity, and temperature. Was there a particular reason you chose these variables to measure. Do they have a relation to habitat type, or environmental tolerances of the local amphibian species? I note you later chose pH over salinity when deciding between correlated variables due to the biological importance of this measure to some species, but some mention of why these measures were chosen in the first place might be beneficial.
Also, was any consideration made to differences in vegetation between the water body types and the role this may play?

4. Line/s: 148-150
Why was the perimeter of the water bodies considered to be more important than the area?

Validity of the findings

no comment

Additional comments

This paper provides important evidence for the consideration of ephemeral wetlands when conserving amphibian species (especially specialised and range restricted species). It is well written and draws clear and ecologically meaningful conclusions. While I do not have personal experience using NMDS, the analyses applied seem to be sound and are well reported in the text. It is therefore my recommendation that it should be accepted for publication with some minor revision (see below). N.B. all line numbers are based on the original “Terblanche-Measey.docx” source file.

In addition to the comments made in sections 1-3 above, I have noted some areas that require minor edits or clarification:

1. Line/s: 33
Keywords missing from manuscript document (although I note that they have been included in the online submission process).

2. Line/s: 50
You mention here that invasive freshwater fish are ranked highly by “all authors” as causing severe impacts. Do you mean all of the authors that you cite in this sentence, or universally in the literature which is how it comes across (which seems unlikely?). Perhaps you can clarify this slightly?

3. Line/s: 52-55
The following sentence: “However, many amphibian communities are driven by natural environmental factors as well as anthropogenically driven creation and modifications of freshwater habitats (Ficetola & De Bernardi, 2004; Hartel et al., 2007; Kruger, Hamer & Du Preez, 2015).” seems a bit awkward here as it breaks up your discussion of invasive fish species impacts and solutions. This sentence might flow better if moved to the end of the first introduction paragraph (i.e. line 46) or to the end of its current paragraph (i.e. line 60).

4. Line/s: 62
Fynbos is a term that is native to South Africa and may be unfamiliar to a more global audience. A brief description of fynbos and the type of environment it encompasses would be hugely beneficial.

5. Line/s: 105
In the following sentence: “The final 50 sites selected encompassed two separate catchments, the spatial proximity and different water body types (see Fig. 1; Supp Info; Table S1).” The “the spatial proximity” part reads a bit awkwardly, and I am not entirely sure what you mean. Could you please reword or clarify this?

6. Line/s: 106
Here you refer to your figure as Fig. 1 but later refer to figures as Figure x.

7. Line/s: 152
Presence absence missing the slash (presence/absence).

8. Line/s: 159-162
It would be useful to list the final variables you used in your models here for clarity (maybe in the brackets).

9. Line/s: 176
You use the number 3 here instead of the word, following the convention of your other reported numbers this should be ‘three’.

10. Line/s: 192-197
Here at the start of the paragraph you mention species determining amphibian community structure (e.g. Xenopus laevis), but isn’t this analysis more indicative of habitat (and the presence of fish) driving species presence/absence (i.e. community structure) with stronger influences on certain species?

11. Line/s: 214-215
Did you happen to find any trends regarding the type of permanent water bodies that the fish inhabited (e.g. did they favour anthropogenically created water bodies over natural etc…)? If so, it might be interesting to note this in your discussion

12. Line/s: 219-223
Do you happen to have any data on the disproportionate number of ponds catering to permanent water vs ephemeral?

13. Line/s: 229
Were any amphibians exclusive to ephemeral habitats?

14. Line/s: 260-262
Are you able to comment on which habitat type these species would be most likely to impact? For example, permanent water bodies or riverine systems?

Reviewer 3 ·

Basic reporting

Overall the text is clear and well written. The manuscript is interesting and woth publishing by PeerJ.
Although the text is concise and sufficient I miss a short description of the different species, their main characteristics, conservation value, distribution area, etc. This could be sumarised in a table and in the M&M. The authors should bear in mind that their reults might be interesting to readers from different parts of the world, so not everybody will be familiar with the species names. I think that will increase the general appeal of the paper.

Experimental design

'no comment'

Validity of the findings

'no comment'

Additional comments

R-squared values from results and Table S2 do not fully match. Please review. I also suggest not to use 4 decimal places in the text, they are not necessary and make it more difficult to follow.

Fig. 3. Please scale properly so all species are visible.

Supplement. Can you please remove unnecessary decimals? At present you show 8 decimals for some results, that are clearly unnecessary. Can you round them a bit so it is easier to read, please?

---

## Round 0.2 · Major Revisions

While the quality of the revised manuscript has significantly improved, and you have addressed most of the comments raised by the reviewers, you still need to implement further changes before your manuscript is accepted for publication. I agree with Reviewer 1 that the methods and details provided on sampling fish are very limited. It is unclear whether you collected fish samples or only relied on the information (images) provided by landowners. If only the latter- you did not collect fish samples, then this is a limited dataset that should not be included in the analyses - unless you provide further details on how you quantified these data for data analysis. Otherwise, provide full details on fish sampling, the data obtained and its analysis.

In addition, you need to address the comments raised by the reviewer (Reviewer 1) on experimental design and statistical analysis/ validity of the findings.

·

Basic reporting

see below

Experimental design

While I appreciate the use of pRDA, regrettably the analysis has some issues. First of all, the data of visit, *in your context*, would only be a confounding variable if you wanted to test for species detectability, which seems not to be the case.
Second, and most important, the spatial predictors that need to be included are in the form of Moran Eigenvector Maps (MEM), which are an eigenfunction method to describe the spatial arrangement of sites. Failing in specify the correct model might have contributed to the small global R2 (10%).
A small detail, the species composition matrix is usually referred to as Y, not X, which should be the environmental predictors.
Third and last, despite the fact that I think you don't have enough evidence to include fish, this could be included in the pRDA as a binary factor (presence or absence of invasive fish species). You surely don't need to run an nMDS neither a PERMANOVA here.

Validity of the findings

I still believe that the evidence for not only fish presence, but also fish species is very scant. Given that the identity of fish is a relevant part of the study, which aims to test the effect of invasive species, only relying on landonwers and pictures is not enough. I'd highly recommend you to drop this analysis from the manuscript.

From Fig. 2, I believe there's some mistakes in your environmental variables matrix (X). You have small dam, large dam, temporary vlei as independent continuous variables, when in fact these should be a single multistate factor, maybe called habitat type. You can't code these as continuous variables, it must be a factor and appear in the ordination diagram as the centroid of the group (see exampled in Legendre. & Legendre 2012 p. 647 Fig. 11.3) The same critique applies to arrows showing temporary/permanent and fish/no-fish. How does these variables, which should be a factor, were coded as continuous? I don't know what river edge means, is that another type of habitat? If so, include it in the first factor I mentioned.

Additional comments

1) Cite Google Earth and the sources of images accordingly
2) Not sure what you intend with Table 1

---

## Round 0.3 · Major Revisions

Reviewer 1 is unsatisfied with your responses to the major comments he raised, which I consider to be very important. You must address these comments fully, especially concerning the fish data and the statistics you have used. Also, note that the reviewer could not reproduce the results using your data. You need to check what is wrong with the analysis and stick to the outputs of the results, as these are different from what you have in the paper.

·

Basic reporting

see below

Experimental design

see below

Validity of the findings

I couldn't reproduce author's results using their data, see attached R markown document compiled in html

---

## Round 0.4 · Minor Revisions

I have gone through your revised manuscript and rebuttal letter, and I am glad to note that the results are reproducible. However, I have not been able to see, in your ms with track changes, the changes you have made in response to the other comments raised by the reviewer. I am keen to see how you have addressed the concerns on fish data and the corresponding statistics. If these data are unavailable, then your discussion should be couched to recognize this shortcoming, and this should also inform the conclusions being drawn. A reviewer can miss something during the first round of review, and, in my opinion, it is proper to raise the matter at any stage of the review process.

---

## Round 0.5 · accepted · Accept

The authors have exhaustively addressed all the outstanding reviewers' comments in the revised manuscript. I have examined the revision myself and I am satisfied with the responses given and the changes made in the manuscript.